# Revisiting inference for ARMA models: Improved fits and superior confidence intervals

**Jesse Wheeler**[1,2]*, **Edward L. Ionides**[1]

**1** Department of Statistics, University of Michigan, Ann Arbor, Michigan, United States of America,
**2** Department of Mathematics and Statistics, Idaho State University, Pocatello, Idaho, United States of America

* jessewheeler@isu.edu

## Abstract

Autoregressive moving average (ARMA) models are widely used for analyzing time series data. However, standard likelihood-based inference methodology for ARMA models has avoidable limitations. We show that currently accepted standards for ARMA likelihood maximization frequently lead to sub-optimal parameter estimates. Existing algorithms have theoretical support, but can result in parameter estimates that correspond to a local optimum. While this possibility has been previously identified, it remains unknown to most users, and no routinely applicable algorithm has been developed to resolve the issue. We introduce a novel random initialization algorithm, designed to take advantage of the structure of the ARMA likelihood function, which overcomes these optimization problems. Additionally, we show that profile likelihoods provide superior confidence intervals to those based on the Fisher information matrix. The efficacy of the proposed methodology is demonstrated through a data analysis example and a series of simulation studies. This work makes a significant contribution to statistical practice by identifying and resolving under-recognized shortcomings of existing procedures that frequently arise in scientific and industrial applications.

## 1 Introduction

Auto-regressive moving average (ARMA) models are the most well known and frequently used approach to modeling time series data. The general ARMA model was first described by Whittle [1], and popularized by Box and Jenkins [2]. Today, ARMA models are a fundamental component of various academic curricula, leading to their widespread use in both academia and industry. ARMA models are as foundational to time series analysis as linear models are to regression analysis, and they are often used in conjunction for regression with ARMA errors. A Google Scholar search for articles from 2024 onward that include the phrase "time series" and the term "ARMA" (or variants) yields over 18,000 results. While not all these articles focus on the same models discussed here, the importance of this model class to modern science cannot be overstated. Given the ubiquity of ARMA models, even small

**Data availability statement:** All data and code relevant to the article are available as supplemental material. Results of the simulation

studies are publicly available at
https://doi.org/10.5281/zenodo.17204366. The
package source code for the arima2 R package
is available at
https://doi.org/10.5281/zenodo.17203988.

**Funding:** The author(s) received no specific
funding for this work.

**Competing interests:** The authors have
declared that no competing interests exist.

improvements in parameter estimation constitute a significant advancement of statistical
practice.

A commonly used extension of the ARMA model is the *integrated* ARMA model, which
extends the class of ARMA models to include first or higher order differences. That is, an
autoregressive integrated moving average (ARIMA) model is an ARMA model fit after differ-
encing the data in order to make the data stationary. Additional extensions include the mod-
eling of seasonal components (SARIMA), or the inclusion of external regressors (SARIMAX).
Our methodology can readily be extended to these model classes as well, but here we focus on
ARMA modeling for simplicity.

We demonstrate that the most commonly used methodologies and software for estimat-
ing ARMA model parameters frequently yield sub-optimal estimates. This assertion may
seem surprising given the extensive application and study of ARMA models over the past
five decades. A natural question arises: if these optimization issues exist, why have they not
been addressed? There are three plausible explanations: the first is that the potential for sub-
optimal results has largely gone unnoticed; the second is a satisfactory solution has not yet
been discovered; and the third is a general indifference to the problem. It is likely that a com-
bination of these factors has deterred prior exploration of this issue. For instance, most prac-
titioners may be unaware of the problem, while those who have noticed it either did not pri-
oritize it or were unable to provide a general method to resolve it. In this article, we address
all three possible explanations by demonstrating the existence of an existing shortcoming,
explaining why the problem has nontrivial consequences, and proposing a readily applicable
and computationally efficient solution.

Imperfect likelihood optimization has an immediate consequence of complicating stan-
dard model selection procedures. Algorithms in widespread use lead to frequent inconsisten-
cies in which a smaller model is found to have a higher maximized likelihood than a larger
model within which it is nested. This is mathematically impossible but occurs in practice
when the likelihood is imperfectly maximized, and is commonly observed using contempo-
rary methods for ARMA models. Such inconsistencies are a distraction for the interpreta-
tion of results even when they do not substantially change the conclusion of the data analysis.
Removing numeric inconsistencies can, and should, increase confidence in the correctness of
statistical inferences.

There are various software implementations available for the estimation of ARMA model
parameters. In this article, we focus on the standard implementations in R (`stats` pack-
age) and Python (`statsmodels` module), which we selected due to their widespread usage.
While both implementations offer multiple ways to estimate parameters, the default approach
in both software packages is to perform likelihood maximization, assuming that the error
terms are Gaussian. The challenges in parameter estimation arise in this situation because
there is no closed-form expression for the likelihood function, though computational algo-
rithms do exist for maximizing ARMA likelihoods [3].

We begin by providing essential background information on the estimation of ARMA
model parameters. We then present our proposed approach for parameter estimation, which
leads to parameter values with a likelihood that is never lower and sometimes higher than the
standard method. This is followed by a motivating example and a discussion of the potential
implications of our proposed method. We also discuss the construction of standard errors for
our maximum likelihood estimate. Specifically, we show that estimates of the standard error
for model parameters that are default output of R and Python can be misleading, and we pro-
vide a reliable alternative. Throughout the article, we use the `stats::arima` function from
the R programming language as a baseline for comparison. The same methodology for fitting

parameters is used in the `statsmodels.tsa` module in Python, and we demonstrate that our results apply to this software as well (S2 Appendix).

## 2 Material and methods

### 2.1 Maximum likelihood for ARMA models

Following the notation of Shumway and Stoffer [4], a time series $\{x_t; t = 0, \pm 1, \pm 2, ...\}$ is said to be ARMA$(p, q)$ if it is (weakly) stationary and

$$x_t = \phi_1 x_{t-1} + \cdots + \phi_p x_{t-p} + w_t + \theta_1 w_{t-1} + ... + \theta_q w_{t-q}, \tag{1}$$

with $\{w_t; t = 0, \pm 1, \pm 2, ...\}$ denoting a mean zero white noise (WN) processes with variance $\sigma_w^2 > 0$, and $\phi_p \neq 0$, $\theta_q \neq 0$. We refer to the positive integers $p$ and $q$ of Eq 1 as the autoregressive (AR) and moving average (MA) orders, respectively. A non-zero intercept could also be added to Eq 1, but for simplicity we assume that the time series $\{x_t; t = 0, \pm 1, \pm 2, ...\}$ has zero mean. We denote the set of all model parameters as $\psi = \{\phi_1, ..., \phi_p, \theta_1, ..., \theta_q, \sigma_w^2\}$. Our objective is to estimate model parameters using the observed uni-variate time series data.

Given the importance of ARMA models, numerous methods have been developed for parameter estimation. For instance, parameters can be estimated using Bayesian inference techniques [5,6], neural networks [7], or by maximizing restricted likelihood [8–10], among other approaches. Specialized methods include those for integer-valued data [11], approaches robust to outliers [12], and non-Gaussian ARMA processes [13]. While each methodology has its merits, we focus on the approach most widely adopted in statistical practice: likelihood maximization, under the assumption that the WN process $\{w_t\}$ is Gaussian.

A relevant method for estimating model parameters is minimization of the conditional sum-of-squares (CSS). The CSS estimate is fast to compute, but it does not posses the statistical efficiency of the maximum likelihood estimate (MLE). However, the CSS method plays a role in likelihood maximization, so we briefly describe it here. By solving for the WN term $w_t$, Eq 1 can be written as

$$w_t = x_t - \sum_{i=1}^{p} \phi_i x_{t-i} - \sum_{j=1}^{q} \theta_j w_{t-j}. \tag{2}$$

A natural estimator would involve minimizing the sum of squares $\sum_{t=1}^{n} w_t^2$. However, since only $x_1, x_2, ..., x_n$ are observed and $w_t$ is recursively defined in Eq 2 using values of $x_{t-p}$ and $w_{t-q}$, directly minimizing this sum is intractable. The CSS method addresses this issue by conditioning on the first $p$ values of the process, assuming $w_p = w_{p-1} = ... = w_{p+1-q} = 0$, and minimizing the conditioned sum $\sum_{t=p+1}^{n} w_t^2$. While the CSS method provides an attractive solution due to its relative simplicity and easiness to compute, it ignores the error terms for the first few observations. This is particularly concerning when the time series is short or when there are missing observations. CSS minimization was previously popular because methods for likelihood maximization were considered prohibitively slow, though this is no longer the case with currently available hardware and software [14].

For likelihood maximization, Eq 1 is reformulated as an equivalent state-space model. Although there are several ways this can be done, the approach of [3] is widely used. In this approach, we let $r = \max(p, q + 1)$ and extend the set of parameters so that $\bar{\psi} = \{\phi_1, ..., \phi_r, \theta_1, ..., \theta_{r-1}, \sigma_w^2\}$, with some of the $\phi_i s$ or $\theta_i s$ being equal to zero unless $p = q + 1$. We define a latent state vector $z_t \in \mathbb{R}^r$, along with transition matrices $T \in \mathbb{R}^{r \times r}$ and $Q \in \mathbb{R}^{r \times 1}$, enabling the recovery of the original sequence $\{x_t\}$ using Eqs 3 and 4. For a detailed explanation of how

to define the latent state $z_t$ and transition matrices $T$ and $Q$ in order to recover the ARMA model, we refer readers to Chapter 3 of Durbin and Koopman [15]. Along with initializations for the mean and variance of $z_0$, these equations allow for the exact computation of the likelihood of the ARMA model via the Kalman filter [16], which can subsequently be optimized by a numeric procedure such as the BFGS algorithm [17].

$$z_t = Tz_{t-1} + Qw_t, \tag{3}$$

$$x_t = \begin{pmatrix} 1 & 0 & ... & 0 \end{pmatrix} z_t. \tag{4}$$

Numeric black-box optimizers require an initial guess for parameter values. For ARMA models, this task is non-trivial because the valid parameter region is defined in terms of the roots of a polynomial associated with the parameters, as discussed in the next section. The default strategy in R and Python is to use the CSS estimator for initialization. This is an effective approach because the CSS estimator asymptotically converges to the MLE [4], and may therefore be close to the global maximum when there are sufficiently many observations. However, the CSS initialization is less useful with limited data, or when there are missing observations. The CSS estimate may also lie outside the valid parameter region, and in such cases, parameters are reinitialized at the origin. Both software implementations also allow for manual selection of initial parameter values, but finding suitable initializations manually can be challenging due to complex parameter inter-dependencies.

The log-likelihood function of ARMA models is often multimodal [14], and therefore this single initialization approach can result in parameter estimates corresponding to local maxima (see Fig 1). This is true even for a carefully chosen initialization, such as the CSS estimate. A common strategy to optimize multimodal loss functions is to perform multiple optimizations using different initial parameters. However, we have found no instances of practitioners using a multiple initialization strategy for estimating ARMA model parameters. This may be explained by a general unawareness of the possibility of converging to a local maximum or because obtaining a suitable collection of initializations for ARMA models is nontrivial. For example, independently initializing parameters at random can place the parameter vector outside the region of interest. Furthermore, uniform random sampling generally fails to adequately cover the plausible parameter region (S1 Appendix).

## 2.2 Novel multi-start algorithms

To obtain random parameter initializations, parameter sets must correspond to *causal* and *invertible* ARMA processes; definitions are in Chapter 3 of Shumway and Stoffer [4]. Let $\{\phi_i\}_{i=1}^p$ and $\{\theta_i\}_{i=1}^q$ be the coefficients of the ARMA$(p,q)$ model (Eq 1), and define $\Phi(x) = 1 - \phi_1 x - \phi_2 x^2 - ... - \phi_p x^p$ as the AR polynomial, and $\Theta(x) = 1 + \theta_1 x + \theta_2 x^2 + ... + \theta_q x^q$ as the MA polynomial. An ARMA model is *causal* and *invertible* if the roots of the AR and MA polynomials lie outside the complex unit circle. As a result, a direct approach to obtaining random parameter initilizations for the target parameter space is to sample the roots of $\Phi(x)$ and $\Theta(x)$ and use these values to reconstruct parameter initializations (Algorithm 1). Alternatively, vectors of length $p$ and $q$ that have elements in the interval $(-1,1)$ can be sampled, and then transformed into the parameter space using the Durbin-Levinson recursion (Algorithm 2), as described in [18].

For the first approach, it is an easier task to sample *inverted* roots, as the sufficient conditions for causality and invertibility now require that the inverted roots lie inside the complex unit circle, a region easier to sample uniformly. The roots of the polynomials can be real or

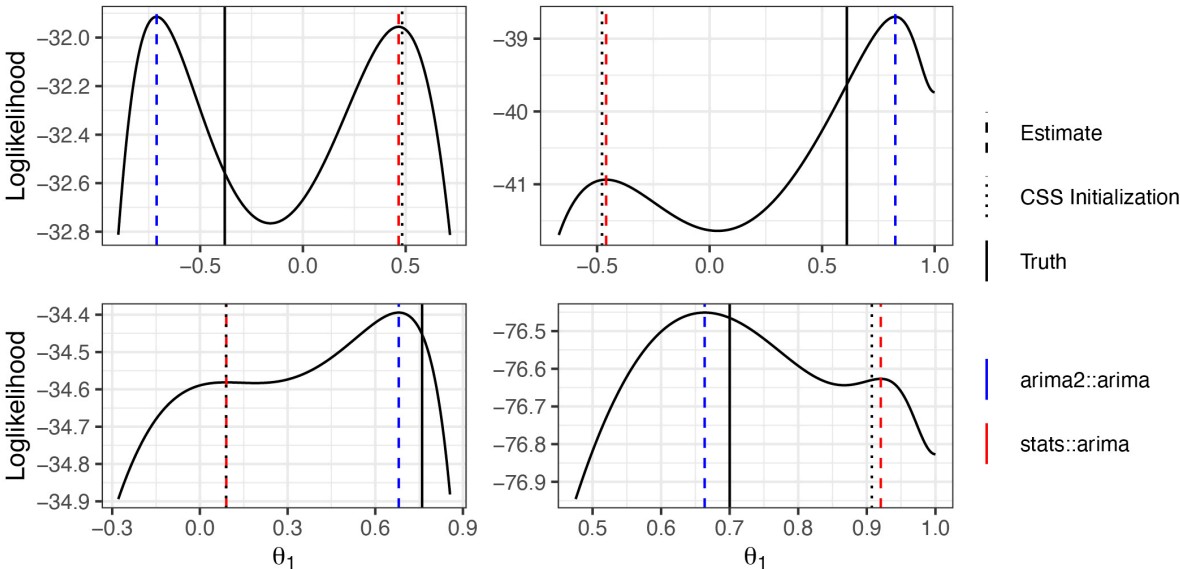

**Fig 1. The profile log-likelihood of data simulated from four distinct MA(1) models, demonstrating a few examples of multimodal likelihood surfaces.** The solid, black line indicates the true value of $\theta_1$; the dotted line is the CSS-initialization. The dashed lines correspond to the estimate $\hat{\theta}_1$ using `stats:arima` (red) and our proposed algorithm (implemented in `arima2::arima`, blue).

**Algorithm 1.   MLE with uniform-root sampling.**
**Inputs (defaults):**
First parameter initialization $\psi_0 = (\phi_1^0, \dots, \phi_p^0, \theta_1^0, \dots, \theta_q^0)$ (CSS estimate).
Minimum acceptable polynomial root distance $\alpha > 0$, ($\alpha = 0.01$).
Bounds on inverted polynomial roots $\gamma \in (0, 0.5)$, ($\gamma = 0.05$).
Numeric optimization routine $f(\psi)$ [3].
Stopping Criterion (stop if last $M$ iterations do not improve log-likelihood $\ell(\psi)$).

```
1  Get preliminary estimate: ψ̂₀ = f(ψ₀); set k = 0;
2  repeat Until stopping criterion met
3       Set AR and MA roots {zᴬᴿ}ᵢ₌₁ᵖ = 0ₚ, {zᴹᴬ}ᵢ₌₁�q = 0q; increment k;
4       while minᵢ,ⱼ |zᵢᴬᴿ − zⱼᴹᴬ| < α, for both AR and MA polynomials  do
5           Sample paired roots as real with probability p = √(1/2);
6           for  all real pairs  do
7               Sample root magnitudes from U(γ, 1 − γ);
8               Sample signs with P(sign(z₁) = sign(z₂)) = p;
9           for  all complex pairs  do
10              sample angle: τ ∼ U(0, π); sample radius: r ∼ U(γ, 1 − γ);
11              set z₁ = r cos(τ) + ir sin(τ); set z₂ = z̄₁;
12          if  Number of roots is odd (non-paired root)  then
13              sample τ uniformly from the set {0, π}; sample r ∼ U(γ, 1 − γ);
14              set z = r cos(τ);
15          Calculate coefficients ψₖ = (φ₁ᵏ, ..., φₚᵏ, θ₁ᵏ, ..., θqᵏ) using sampled roots;
16          Estimate ψ̂ₖ = f(ψₖ);
17  until ;
18  Set ψ̂ = arg max_{j∈0:k} ℓ(ψ̂ⱼ);
```

complex; complex roots must come in complex conjugate pairs in order for all of the corresponding model parameters to be real. The simplest approach would be to sample all inverted root pairs $(z_1, z_2)$ within the complex unit circle by uniformly sampling angles and radii (lines 9-14 of Algorithm 1). However, this would imply almost surely all root pairs are complex, and some model parameters would only be sampled as positive (or negative). For instance,

consider an AR(2) model. The AR polynomial is:

$$\Phi(x) = 1 - \phi_1 x - \phi_2 x^2 = (1 - z_1 x)(1 - z_2 x) = 1 - (z_1 + z_2)x - (z_1 z_2)x^2$$

In this equation, if both $z_1, z_2$ are complex conjugates, then $\phi_2 = z_1 z_2 > 0$. As such, the only way that $\phi_2 < 0$ is if $z_1, z_2 \in \mathbb{R}$. Similar results hold for the MA coefficients with opposite signs for the coefficients. This issue is directly addressed in lines 5-8 of Algorithm 1: root pairs are sampled as real with probability $p = \sqrt{1/2}$, and real pairs sampled with the same sign with probability $p$, such that the product (and sums) of each pair is positive with probability 1/2. We sample conjugate pairs within an annular disk on the complex plane to avoid trivial and approximately non-stationary cases. The radii of both the inner and outer circles defining the disk are defined using $\gamma$ in lines 7, 10, and 13 of Algorithm 1.

Algorithm 1 is designed to directly sample the roots of the AR and MA polynomials, as the roots determine the causality and invertibility of the resulting model. However, it is not immediately evident how the choice of sampling a proportion $p = \sqrt{1/2}$ will effect the final distribution of ARMA coefficients, other than ensuring that parameters are sampled as positive or negative with similar frequencies. Existing theory can be used to provide an alternative approach to sampling model parameters within the admissibility region [18]. Algorithm 2 leverages existing techniques by first generating a vector of partial autocorrelations, each taking values in the interval (–1,1). These vectors are transformed into ARMA parameters using the Durbin-Levinson recursion, ensuring the resulting parameters are within the admissible region. This process is outlined in Eqs (4) and (5) of [10], who demonstrate that the established theory for this transformation continues to hold at boundary conditions.

**Algorithm 2.   MLE with Durbin-Levinson sampling**.
**Inputs (defaults):**
First parameter initialization $\psi_0 = (\phi_1^0, \ldots, \phi_p^0, \theta_1^0, \ldots, \theta_q^0)$ (CSS estimate).
Minimum acceptable polynomial root distance $\alpha > 0$, ($\alpha = 0.01$).
Numeric optimization routine $f(\psi)$ [3]).
Stopping Criterion (stop if last $M$ iterations do not improve log-likelihood $\ell(\psi)$).

```
 1  Get preliminary estimate: ψ̂₀ = f(ψ₀); set k = 0;
 2  repeat Until stopping criterion met
 3      Increment k;
 4      if  p (or q) = 1  then
 5          Sample  φ₁ (or θ₁) uniformly from U(−1 + γ, 1 − γ);
 6          Set corresponding root z₁ᴬᴿ (or z₁ᴹᴬ) to be  φ₁ (or θ₁);
 7      else
 8          Set AR and MA roots {zᵢᴬᴿ}ᵢ₌₁ᵖ = 0ₚ,  {zⱼᴹᴬ}ⱼ₌₁ᵠ = 0_q;
 9      Increment k;
10      while  minᵢ,ⱼ |zᵢᴬᴿ − zⱼᴹᴬ| < α  do
11          Uniformly sample  ξᵢ,ᵢᴬᴿ, ξⱼ,ⱼᴹᴬ ~ⁱ·ⁱ·ᵈ· U(γ, 1 − γ) for i ∈ 1:p, j ∈ 1:q;
12          for  i in 2:p  do
13              for  h in 1:(i − 1)  do
14                  ξᵢ,ₕᴬᴿ = ξᵢ₋₁,ₕᴬᴿ − ξᵢ,ᵢᴬᴿ ξᵢ₋₁,ᵢ₋ₕᴬᴿ;
15          for  j in 2:q  do
16              for  h in 1:(j − 1)  do
17                  ξⱼ,ₕᴹᴬ = ξⱼ₋₁,ₕᴹᴬ − ξⱼ,ⱼᴹᴬ ξⱼ₋₁,ⱼ₋ₕᴹᴬ;
18          Set  ψₖ = (φ₁ᵏ, ..., φₚᵏ, θ₁ᵏ, ..., θ_qᵏ) = (ξ_{p,1}ᴬᴿ, ..., ξ_{p,p}ᴬᴿ, −ξ_{q,1}ᴹᴬ, ..., −ξ_{q,q}ᴹᴬ);
19          Calculate roots {zⱼᴬᴿ}ⱼ₌₁ᵖ, {zⱼᴹᴬ}ⱼ₌₁ᵠ from sampled coefficients;
20      Estimate  ψ̂ₖ = f(ψₖ);
21  until;
22  Set  ψ̂ = arg max_{j∈0:k} ℓ(ψ̂ⱼ);
```

Parameter redundancy in ARMA models occurs when the polynomials $\Phi(x)$ and $\Theta(x)$ share one or more roots, leading to an overall reduction in model order. This complicates parameter initialization, optimization, and identifiability. The ARMA model (Eq 1) can be rewritten as:

$$\Phi(B)x_t = \Theta(B)w_t, \qquad (5)$$

where $B$ is the *backshift* operator, i.e., $Bx_t = x_{t-1}$. Using the fundamental theorem of algebra, Equation 5 can be factored into

$$(1 - \lambda_1 B) \dots (1 - \lambda_p B)x_t = (1 - \nu_1 B) \dots (1 - \nu_q B)w_t,$$

where $\{\lambda_i\}_{i=1}^p$ and $\{\nu_j\}_{j=1}^q$ are the inverted roots of $\Phi(B)$ and $\Theta(B)$, respectively. If $\lambda_i = \nu_j$ for any $(i, j) \in \{1, \dots, p\} \times \{1, \dots, q\}$, then the roots will cancel each other out, resulting in an ARMA model of smaller order. As an elementary example, consider the ARMA$(1, 1)$ and ARMA$(2, 2)$ models in Eqs 6 and 7.

$$x_t = \frac{1}{3}x_{t-1} + w_t + \frac{2}{3}w_{t-1}, \qquad (6)$$

$$x_t = \frac{5}{6}x_{t-1} - \frac{1}{6}x_{t-2} + w_t + \frac{1}{6}w_{t-1} - \frac{1}{3}w_{t-2}. \qquad (7)$$

While these two models appear distinct at first glance, re-writing the models in polynomial form (Eq 5) shows that these two models are actually equivalent after canceling out the common factors on each side of the equation.

In a similar fashion, it is possible that the roots are not exactly equal but are approximately equal. In this case, the ratio of factors becomes close to one, resulting in a similar effect to when the roots exactly cancel. In Algorithms 1 and 2, we avoid the possibility of *nearly canceling roots* in parameter initializations by requiring the minimum Euclidean distance between inverted polynomial roots to be greater than $\alpha$. This is done in line 4 of Algorithm 1 and line 10 of Algorithm 2, though the condition is rarely triggered if the order of the model is of typical size ($p, q < 4$).

Both of the sampling schemes are combined with existing procedures for numeric optimization of model log-likelihoods (lines 16 and 20, respectively), as well as the default initialization strategy for $\psi_0$ used by existing software (such as the CSS initialization, in line 1 of both algorithms). Doing so guarantees that final estimates obtained from both algorithms correspond to likelihood values greater than or equal to the currently accepted standards in the software environment where the algorithms are implemented. For this article, both the numeric optimization procedure $f(\cdot)$ and the parameter initialization strategy to obtain $\psi_0$ are those implemented in `stats::arima`. The stopping criterion was chosen so that the algorithm stops trying new initial values when no new maximum has been found using the last $M$ parameter initializations. Alternative stopping criterion can be used (see for example, [19]), but we found that this simple heuristic works well in practice.

Both algorithms are implemented in the R package `arima2`, available on the Comprehensive R Archive Network (CRAN) [20]. The package features the function `arima2::arima`, which is an adaptation of the `stats::arima` function modified to incorporate the new initialization schemes. The choice of initialization scheme can be selected using the `init_method` argument to the function.

## 3 Results

### 3.1 Simulation studies

To investigate the extent to which the standard approach for ARMA parameter estimation results in improperly maximized likelihoods, we conduct a series of simulation studies. It is challenging to obtain precise estimates of how frequently current standards lead to sub-optimal parameter estimates due to the varied applications of ARMA models in practice, the diversity in data sizes ($n$) and model orders ($p,q$), and the differing degrees to which an ARMA model adequately describes the data-generating process. Therefore, we restrict our simulation studies to idealized scenarios where the data-generating process is Gaussian-ARMA, recognizing that likelihood maximization is easiest for this model class, thereby resulting in conservative estimates of how frequently our algorithm improves model likelihood.

In the first simulation study, we simulate time series data of lengths $n \in \{50, 100, 500, 1000\}$ from Gaussian-ARMA models with known orders $(p, q) \in \{1, 2, 3\}^2$. For each combination of (p, q, n), 1000 unique parameter sets are generated, as well as a single time series from each set of parameters. In total, this results in 36,000 unique models and datasets. We avoid any models that contain parameter redundancies by requiring the data generating model to have a minimum distance of 0.1 between all roots of $\Phi(x)$ and $\Theta(x)$. We further restrict model coefficients so that they do not lie near boundary conditions. Models of the same order of the generating data are fit to the data, simplifying the the problem further by avoiding the order selection step that is necessary in most data analyses. In doing so, we attempt to answer the question of how often sub-optimal estimates may arise using existing software in the case where the parameter estimation procedure should be as easy as possible for the given combinations of ($n$, $p$, $q$).

Even in this extremely simplified scenario, existing software failed to properly maximize model likelihoods in at least 23.4% of the simulated datasets—evidenced by an improvement obtained using either Algorithm 1 or Algorithm 2. Though this improvement may appear modest, an improvement in 23.4% of the large number of published ARMA models would affect many papers—a number measured in thousands of papers since 2024 following our estimate in the introduction. Furthermore, time series analysis courses and textbooks often recommend fitting multiple ARMA models to a dataset, and here we only fit one for each algorithm. Consequently, the probability that at least one candidate model is not properly optimized increases significantly in practice. The rate of improvement obtained using our algorithm increases with model complexity and decreases with more observations (Fig 2). For example, likelihoods improved in 61.9% of the simulations when $p = q = 3$ and $n = 50$ by using both algorithms. Models of this size and number of observations are not uncommon in published research studies.

In this simulation study, we found that Algorithm 1 generally resulted in greater improvements over the baseline compared to Algorithm 2. Specifically, Algorithm 1 improved the log-likelihood in 20.7% of the simulated data relative to the baseline, whereas Algorithm 2 did so in 20.4% of cases. Moreover, when the two algorithms produced different outputs, Algorithm 1 more often achieved the highest log-likelihood. For simplicity, the remainder of the article summarizes the results obtained using only Algorithm 1 rather than both algorithms. However, the fact that some improvements were achieved by Algorithm 2 alone indicates that it may be more suitable for certain datasets, or that increasing the number of random initializations considered by having a more strict stopping criteria may be beneficial.

Importantly, we do not claim to improve likelihood for all ARMA models and datasets. In this simulation study, our likelihood maximum routine did not improve likelihoods for many

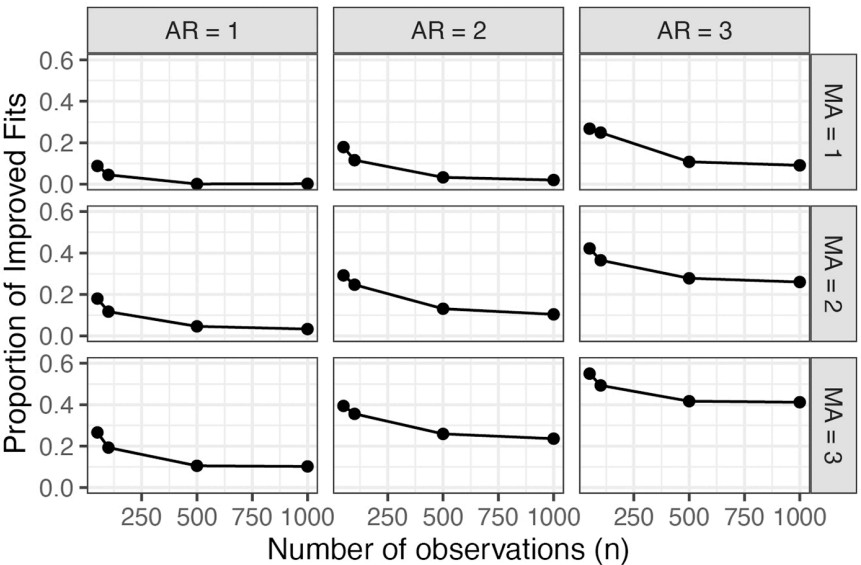

**Fig 2. Proportion of simulated data with improved likelihood from using multiple restarts (Algorithm 1).**

of the simulated data. However, these results demonstrate that there is a very real potential for obtaining sub-optimal parameter estimates when using only a single parameter initialization, even in the most idealistic scenarios. Rather than having to worry if a single initialization is sufficient to fit a given model, it is preferable to adopt methods that make such situations rare. The primary limitation of our algorithm is that the potential for improved fits comes at the cost of increased computation times. In our simulation study, however, the average time to estimate parameters using our approach was 0.6 seconds, a computational expense that is worth the effort in many situations. Average computing times from our simulation study are given in Table 1. The amount of computing time required for each time series depends heavily on many factors, including the proximity of parameter initializations to local optimums. When $n$ is large, the CSS initialization will be closer to one of these local optimums, resulting in much faster convergence than randomly initialized parameters.

The median log-likelihood improvement in this simulation study was 0.66, with an interquartile range of (0.22,1.47). Among the most common motivations for fitting ARMA models is to model serial correlations in a regression model; in this setting, the discovered shortcomings in log-likelihood are often enough to change the outcome of the analysis. For instance, consider modeling $y_i = \beta x_i + \epsilon_i$, where $\beta \in \mathbb{R}$, and we model the error terms

**Table 1. Table summarizing the computing times of each simulated dataset, in seconds.** For large $n$, the CSS initialization is generally close to a maximum, causing the default initialization to converge quicker than random initializations.

| (a) Algorithm 1 | | | (b) Algorithm 2 | | | (c) `stats::arima` | | |
|---|---|---|---|---|---|---|---|---|
| $n$ | Mean | SD | $n$ | Mean | SD | $n$ | Mean | SD |
| 50 | 0.28 | 0.20 | 50 | 0.30 | 0.21 | 50 | 0.03 | 0.02 |
| 100 | 0.33 | 0.24 | 100 | 0.35 | 0.25 | 100 | 0.03 | 0.02 |
| 500 | 0.74 | 0.58 | 500 | 0.78 | 0.60 | 500 | 0.05 | 0.04 |
| 1000 | 1.13 | 0.90 | 1000 | 1.20 | 0.92 | 1000 | 0.07 | 0.06 |

$\epsilon_i \sim \text{ARMA}(p,q)$. For now, we will assume the order $(p,q)$ is fixed. We may wish to test the hypothesis $H_0 : \beta = 0$ vs $H_1 : \beta \neq 0$. A standard approach to doing this is a likelihood ratio test, and using Wilks' theorem to get an approximate test. We denote $ll_0$ and $ll_1$ as the maximum log-likelihood of the model under $H_0$ and $H_1$, respectively. The standard approximation is to assume $2\Delta = 2(ll_1 - ll_0) \sim \chi_1^2$. Using a significance level of $\alpha = 0.05$, we would reject $H_0$ if $\Delta \geq 1.92$. Given that $E_{H_0}[\Delta] = 0.5$, subtracting the reported log-likelihood deficiencies (which has a median value of 0.66) of existing software to either or both $ll_0, ll_1$ could change the outcome of this test.

Likelihood optimization for mixed ARMA models is known to be more challenging than for pure AR or MA processes. Nonetheless, Fig 1 demonstrates that multimodal likelihood surfaces and suboptimal estimates can arise in pure MA(1) processes, and even these simple models can benefit from multiple initializations. This class of models is particularly useful for illustrating the potential for suboptimal estimates, as the presence of only a single parameter allows for easy visualization. Our preliminary results indicate, however, that pure AR and MA processes rarely require multiple initializations for effective optimization. The difficulty of parameter optimization in mixed ARMA models is one reason authors have previously advised caution in their use [21]. Despite the need for caution, mixed models remain commonly used in practice and are therefore the primary focus of our article.

**3.1.1 Parameter uncertainty.** Improved parameter estimation leads to modified standard error estimates, which are default outputs in R and Python. These standard errors result from the numeric optimizer's estimate of the gradient of the log-likelihood, used to approximate the Fisher information matrix. These standard errors, though not inherently of interest, are sometimes used justify the inclusion of a parameter in a model [22, Chapter 9]. We extend our simulation study to examine this approach and how our algorithm impacts the estimates. For each of the 36,000 generative models from the previous study, we generate 100 additional datasets, estimating the MLE and Bonferroni-adjusted 95% confidence intervals using both estimated standard errors and profile likelihood confidence intervals (PLCIs) from Wilks' theorem. The accuracy of confidence intervals is evaluated by comparing the nominal coverage of the confidence intervals from the simulations to the target 95% coverage level. Fig 3 shows PLCIs had better or equivalent nominal coverage than Fisher-based confidence intervals across all combinations of $p$, $q$, and $n$. We found that the interval estimation method mattered more for confidence interval performance than the specific parameter estimation algorithm. The relevance of this result is explored further in Sect 3.2.1.

**3.1.2 AIC table consistency.** A more realistic situation than the previous simulation study involves estimating the model order $(p,q)$ as well as obtaining parameter estimates. Fitting multiple models raises the chance that at least one candidate model was not properly optimized. It may also necessitate fitting larger models than needed, leading to parameter redundancies that make proper optimization more challenging.

A contemporary approach involves fitting several candidate models and selecting the one that minimizes a criterion like Akaike's information criterion (AIC) [23]. This can be done by explicitly creating a table of all candidate models and their corresponding AIC values; in this case issues of improper maximization become more apparent. For instance, a table of AIC values may contain numeric inconsistencies, where a larger model may have lower estimated likelihoods than a smaller model within which it is nested (for an example, see Sect 3.2). This type of result can make a careful practitioner feel uneasy, as there is evidence that at least one candidate model was not properly optimized. Evidence of improper optimization may be less evident when relying on software that automates this process, such as the automated Hyndman-Khandakar algorithm [24], but the potential for sub-optimal estimates remains.

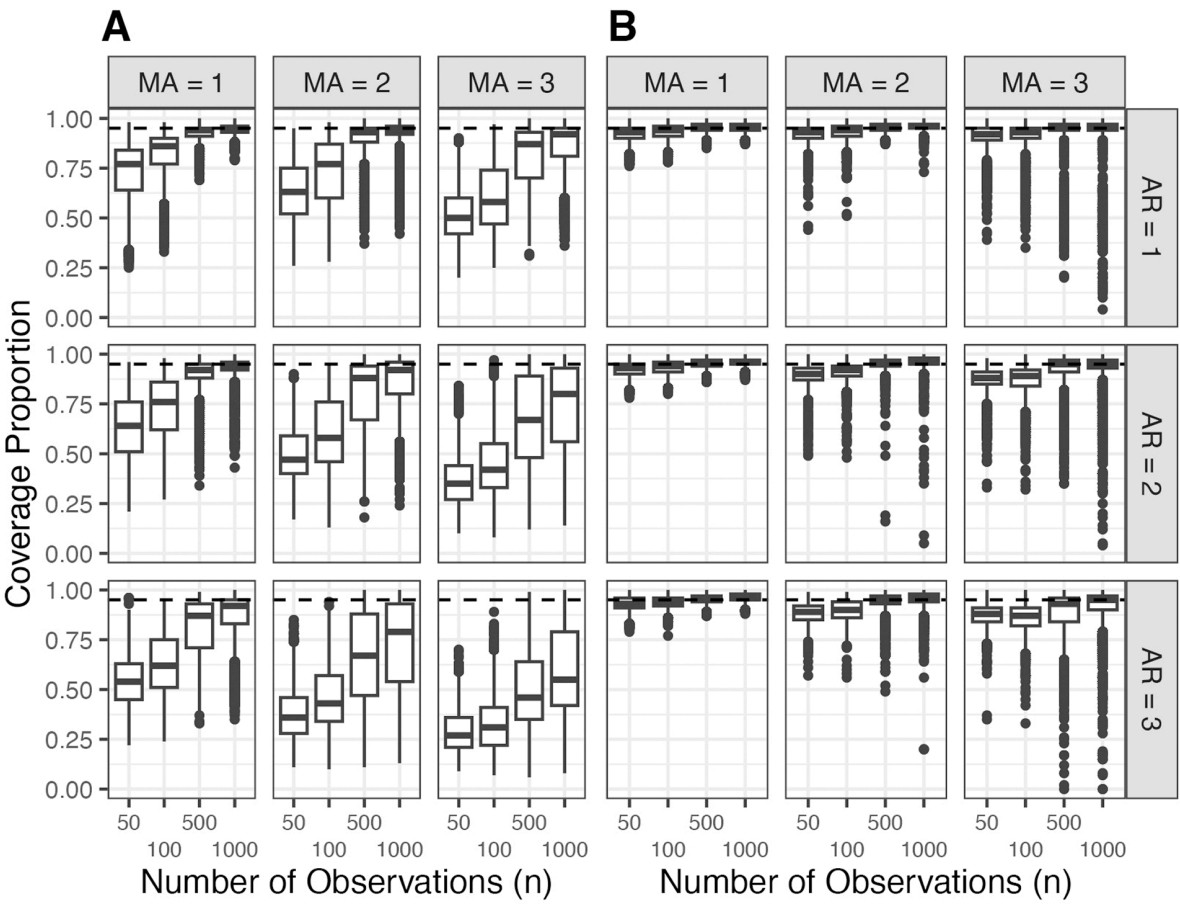

**Fig 3. Proportion of models that achieved nominal coverage of Bonferroni adjusted 95% confidence intervals.** The dashed line denotes the target coverage level. Parameter estimates were obtained using Algorithm 1. (A) Confidence intervals created using Fisher's information matrix. (B) Confidence intervals created using profile likelihoods.

We conducted an additional simulation study to investigate numeric inconsistencies that may arise when fitting multiple model parameters. As before, we simulated 1000 unique models and datasets of size $n \in \{50, 100, 500, 1000\}$ from Gaussian ARMA$(p,q)$ models for $(p, q) \in \{1, 2, 3\}^2$. To avoid models with parameter redundancies, we ensured a minimum distance of 0.1 between all roots of $\Phi(x)$ and $\Theta(x)$ and excluded models with coefficients near boundary conditions. For each dataset, AIC tables were created for model sizes $(p, q) \in \{0, 1, 2, 3\}^2$.

The single parameter initialization approach resulted in AIC table inconsistencies in 45.6% of the simulated datasets. Although our proposed algorithm significantly mitigates this issue, it does not guarantee that all model likelihoods are fully maximized. This is illustrated in Fig 4, where a non-zero percentage of AIC tables remain inconsistent, even as the algorithm's stopping criterion grows. The ARMA(1,1) panel in Fig 4 illustrates the increasing difficulty of parameter estimation when dealing with parameter redundancies. In such cases, it is often necessary to adjust additional parameters in the numeric optimization routine. For example, our R implementation of the algorithm relies on the generic BFGS optimizer in the `stats::optim` function. Modifying the optimization method or the default hyperparameters can lead to improved fits or faster convergence rates. While the default parameters of the

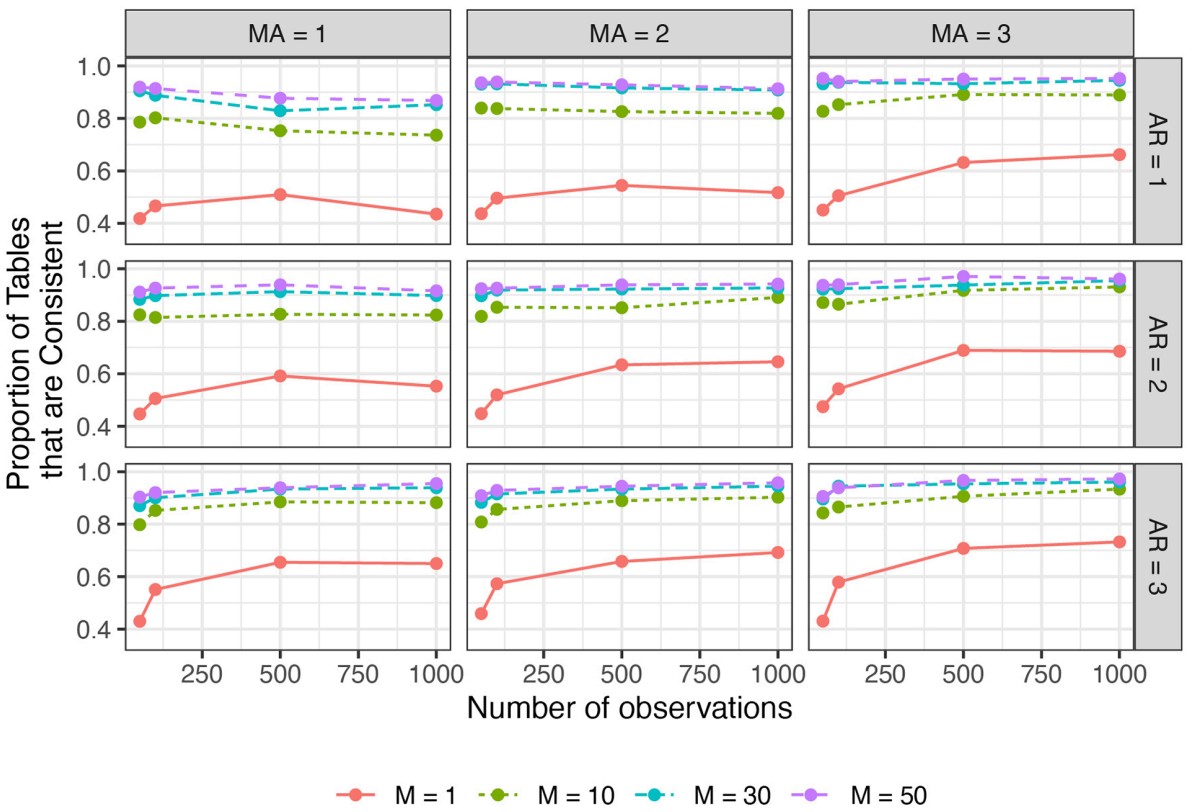

**Fig 4. Data is generated from ARMA $(p, q)$ models with $(p, q) \in \{1, 2, 3\}^2$, and the corresponding AIC table is created.** The Y-axis shows the percentage of tables that were consistent. M is the number of times a maxima is observed before the algorithm terminates, so $M = 1$ corresponds to the standard maximization procedure.

numeric optimizer are generally adequate, increasing the maximum number of algorithmic iterations can be beneficial for fully maximizing the likelihood for challenging models and data.

The contemporary approach of using AIC—or any other information based criteria—to select model order involves fitting unnecessarily large models, leading to parameter redundancies that complicate likelihood optimization. The use of AIC for ARMA model order selection has theoretical support, particularly for forecasting, as ARMA models inspired the original AIC paper [23]. However, without proper likelihood maximization, a strategy that considers only a single parameter initialization may not truly minimize AIC. In this framework, likelihood maximization and over-parameterization are interconnected: all candidate models must be maximized for likelihood, or users risk selecting over-parameterized models that fail to minimize the intended information criterion. For the current study, the choice of AIC versus other popular information criteria such as the corrected AIC (AICC) or Bayesian information criterion (BIC) is unimportant: all of these approaches rely on proper optimization of the likelihood function, which is the problem we are addressing here.

Classical ARMA modeling addresses this by recommending diagnostic plots to determine appropriate model order and advising against simultaneously adding AR and MA components. In this approach, the additional difficulty in parameter estimation associated with fitting models containing parameter redundancies is avoided by not fitting overly complex models when possible. Despite this, shortcomings in likelihood maximization can occur

even in models without parameter redundancies (Figs 1, 2, and 4), necessitating the exploration of multiple parameter initializations. Further, the increasing preference of using automated software to pick the model size using an information criterion suggests the importance of using software that reliably maximizes model likelihoods even in the presence of over-parameterization.

## 3.2 Annual depths of lake Michigan

In this example, we illustrate how improperly maximized likelihoods can lead to inconsistencies and uncertainty in a real data analysis scenario. Additionally, we show how the common practice of using the estimated standard error for calibrated parameters can misleadingly support the inclusion of model parameters. We consider a dataset containing annual observations on the average depth of Lake Michigan-Huron, recorded the first day of each year from 1860-2014 (Fig 5) [25]. The data are provided as part of the `arima2` package. We wish to develop an ARMA model for these data, which is a standard task in time series analysis [4].

Diagnostic tests, such as sample autocorrelation and normal quantile plots for residuals, suggest that it is reasonable to model the data in Fig 5 as a weakly stationary Gaussian ARMA$(p, q)$ process for some non-negative integers $p$ and $q$. While an ARIMA model may also be reasonable for these data, we first consider fitting an ARMA model because we would like to avoid the possibility of over-differencing the data. The next step is to determine appropriate values of $p$ and $q$; after some initial investigation, multiple combinations of $p$ and $q$ seem plausible, and therefore we decide to choose the values of $p$ and $q$ that minimize the AIC. For simplicity, we create a table of AIC values for all possible combinations of $(p, q) \in \{0, 1, 2, 3\}^2$ (Table 2). Using the AIC as the model selection criterion, the selected model size is ARMA$(2, 1)$.

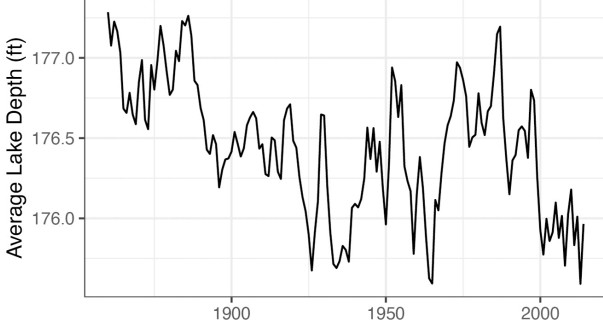

**Fig 5. Average depth of Lake Michigan-Huron from 1860-2014.**

**Table 2. AIC values for an ARMA(p, q) model fit to Lake Michigan-Huron depths.** Table 2a was computing using only a single parameter initialization. Table 2b was computed using Algorithm 1. Highlighted cells show where the likelihood was improved (AIC reduced) using our algorithm.

|     | MA0   | MA1   | MA2   | MA3   |     | MA0   | MA1   | MA2   | MA3   |
|-----|-------|-------|-------|-------|-----|-------|-------|-------|-------|
| AR0 | 166.8 | 46.6  | 7.3   | −15.0 | AR0 | 166.8 | 46.6  | 7.3   | −15.0 |
| AR1 | −38.0 | −37.4 | −35.5 | −33.8 | AR1 | −38.0 | −37.4 | −35.5 | −33.8 |
| AR2 | −37.3 | −38.4 | −36.9 | −34.9 | AR2 | −37.3 | −38.4 | −36.9 | −34.9 |
| AR3 | −35.5 | −35.2 | −33.0 | −33.4 | AR3 | −35.5 | −36.9 | −36.4 | −34.2 |

Recall that the AIC is defined as:

$$\text{AIC} = -2 \max_{\psi} \ell(\psi; x^*) + 2d, \tag{8}$$

where $\ell(\psi; x^*)$ denotes the log-likelihood of a model indexed by parameter vector $\psi \in \mathbb{R}^d$, $d \geq 1$, given the observed data $x^*$. In the case of an ARMA model with an intercept, $d = p + q + 2$, where the additional parameter corresponds to a variance estimate. If either $p$ or $q$ increases by one, then a corresponding increase in AIC values greater than two suggests that the *inclusion* of an additional parameter resulted in a *decrease* in the maximum of the log-likelihood, which is mathematically impossible under proper optimization. Several such cases are present in Table 2a, for example increasing from an ARMA$(2, 2)$ model to a ARMA$(3, 2)$ model results in a decrease of 1.0 log-likelihood units. In this case, using our multiple restart algorithm eliminates all instances of mathematical inconsistencies (Table 2b). We refer to tables that have log-likelihood values larger for any smaller nested model within the table as *inconsistent*.

Suppose a scientist is confronted with a mathematically implausible table of nominally maximized likelihoods (Table 2a). How much should they worry about this? Is it acceptable to publish scientific results that demonstrate a nominally maximized likelihood is not, in fact, maximized? Can researchers confidently trust the scientific implications of a fitted model if there is evidence of improper optimization in some of the candidate models? Given a choice, a researcher should prefer to use maximization algorithms reliable enough to make such situations rare. In the Lake Michigan example, improved estimation does not change which model is selected or the final parameter estimates, but it does remove inconsistencies that could lead to these concerns (Table 2b).

Minimizing the AIC (or an alternative information criterion) is not the only accepted approach to order selection. A classical perspective on model selection involves consulting sample autocorrelation plots, partial autocorrelation plots, conducting tests such as Ljung-Box over various lags, studying the polynomial roots of fitted models, and checking properties of the residuals of the fitted models [2,4,22]. This approach helps avoid fitting models that are possibly over-parameterized. However, additional computational power and increasing volumes of data have favored automated data analysis strategies that fit many models and evaluate them using a model selection criterion. In principle, a simple model selection criterion such as AIC can address parsimony and guard against over-parameterization as well. Diagnostic inspection can be combined with these automated approaches. For example, a table of AIC values can be generated, and models with promising likelihoods can be explored further [22].

When possible, there may be general agreement that the best approach is to combine modern computational resources with careful attention to model diagnostics, considering the data and the scientific task at hand. Improved maximization facilitates this process by eliminating distractions resulting from incomplete maximization.

**3.2.1 Parameter uncertainty.** Default output from fitting an ARMA model in R or Python includes estimates for parameter values and their standard errors, calculated using Fisher's information matrix. If the ARMA model with the lowest AIC value is chosen to describe the Lake Michigan data, then an ARMA$(2, 1)$ model is selected. The estimated coefficients and standard errors obtained after fitting this model are reported in Table 3. The small standard error for $\hat{\theta}_1$ reported in this table suggests a high-level of confidence that the parameter has a value near 1. Taken at face value, these estimates seem to strongly favor the inclusion of the MA(1) term in the model.

**Table 3. Parameter values of ARMA($p, q$) model fit to Lake Michigan-Huron depth data.** The same parameters are returned with a single initialization, or when using Algorithms 1 or 2.

| | $\phi_1$ | $\phi_2$ | $\theta_1$ | Intercept |
|---|---|---|---|---|
| Estimate | -0.053 | 0.791 | 1.000 | 176.460 |
| s.e. | 0.052 | 0.053 | 0.024 | 0.121 |

However, our simulation studies have suggested that these confidence intervals can be misleading, and that PLCIs are more reliable alternatives. The 95% PLCI for the parameter (Fig 6A) is much larger than the confidence interval created using these standard errors. The steep curve in the immediate vicinity of $\hat{\theta}_1$ may explain the small standard error estimates for this parameter and the corresponding tight confidence intervals created using Fisher's identity matrix. Alternative evidence indicates the potential for nearly canceling roots (Fig 7), in which case the MA(1) term may not be needed in the model.

Both types of confidence intervals considered in this example rely on asymptotic justifications, but we can further investigate the finite sample properties using a simulation study. We fit both ARMA(2, 1) and AR(1) models to the data, and conduct a boot-strap simulation study by simulating 1000 datasets from each of the fitted models. We then re-estimate an ARMA(2, 1) model to each of these datasets and record the estimated coefficients. A histogram containing the estimated values of $\hat{\theta}_1$ when the data are generated from the ARMA(2, 1) and AR(1) models fit to the Lake Huron-Michigan data are shown in Fig 6B and Fig 6C, respectively.

The shape of this histogram in Fig 6B mimics that of the profile log-likelihood surface in Fig 6A, confirming that a large confidence interval is needed in order to obtain a 95% confidence interval. In Fig 6C, a large number of $\hat{\theta}_1$ coefficients are estimated near 1 when the generating model is AR(1). Combining this result with the nearly canceling roots of the ARMA(2, 1) model (Fig 7), we cannot reject the hypothesis that the data were generated from a AR(1) model, even though the Fisher information standard errors suggest that the

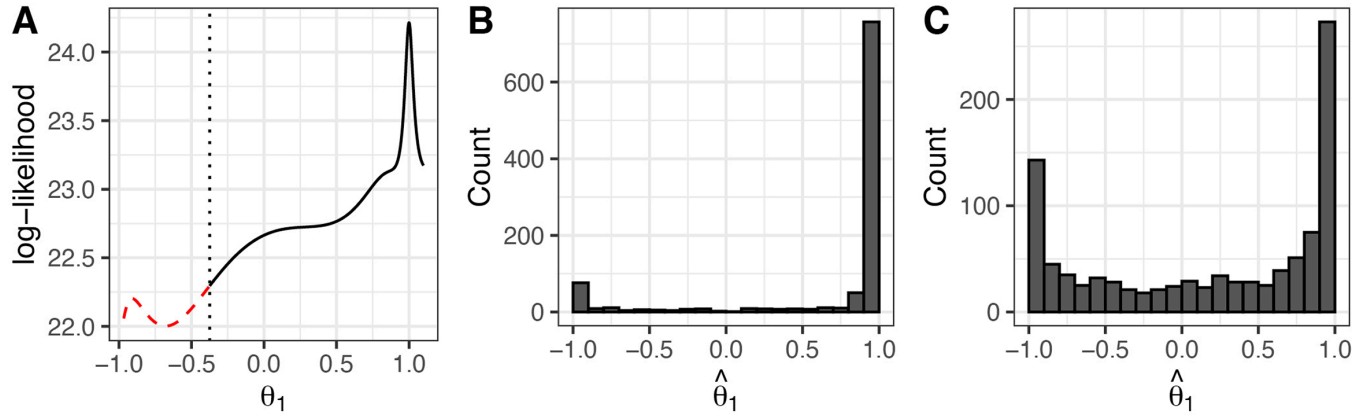

**Fig 6. Evidence for an AR(1) model for the Lake Michigan-Huron data.** (A) Profile likelihood confidence interval (PLCI) for $\theta_1$ which includes the value $\theta_1 = 0$. The vertical dotted line represents the lower end of the approximate confidence interval; all points on the solid black line lie within the confidence interval, and points on the dashed red line are outside the interval. (B) Histogram of re-estimated $\theta_1$ values using simulated data simulated from the ARMA(2, 1) model that was calibrated to the Lake Michigan-Huron data. (C) Histogram of re-estimated $\theta_1$ values using data simulated from the AR(1) model that was calibrated to the Lake Michigan-Huron data.

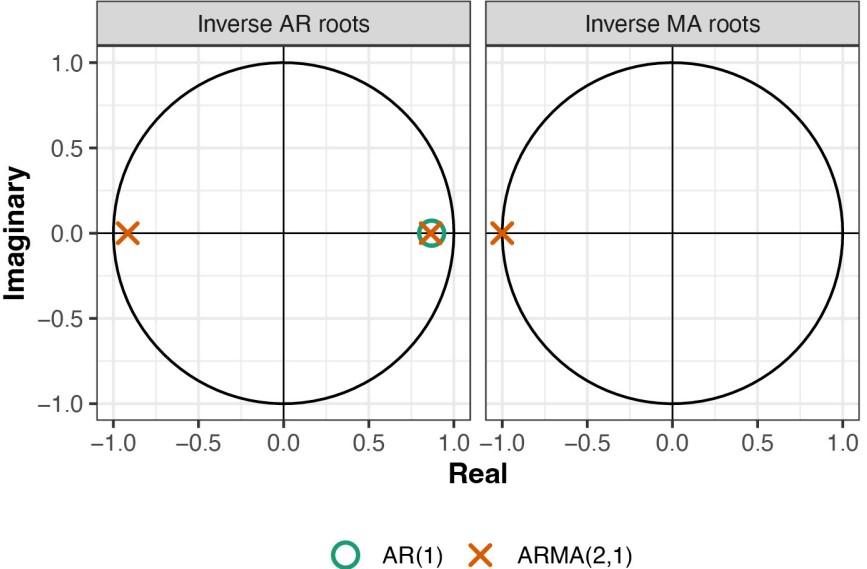

**Fig 7. Inverted AR and MA polynomial roots to the fitted AR(1) and ARMA(2, 1) models to the Lake Michigan-Huron data using a single parameter initialization.**

data should be modeled with a nonzero $\theta_1$ coefficient. Diagnostic plots for both AR(1) and ARMA(2, 1) models are given in Fig 8.

## 4 Discussion

A significant motivation for our work is the observation that commonly used statistical software that purports to maximize ARMA model likelihoods fails to do so for a large number of examples. In addition to improving parameter estimates, proper maximization of ARMA model likelihoods is crucial because ARMA models are often used to model serial correlations in regression analyses. In this context, researchers may perform likelihood ratio hypothesis tests for regression coefficients, and the validity of these tests depends on proper likelihood optimization.

An important consequence of improved likelihood maximization is better model selection. A common approach to selecting an ARMA model involves fitting different sizes of models and choosing the one that minimizes an information criterion, such as the AIC. Fitting multiple models results in having a higher probability that at least one candidate model was not properly maximized. Since AIC assumes the parameters correspond to maximized likelihoods, enhancements in likelihood maximization can lead to different model selections. Consequently, methodology relying on existing estimation methods—like the popular `auto.arima` function in the `forecast` package in R [24], which minimizes the AIC of a group of candidate models without explicitly displaying an AIC table—will be impacted by improved estimates.

Our proposed algorithm is supported by existing theory on likelihood evaluation of linear state-space models via the Kalman Filter [16], the same as the current existing standard approach for parameter estimation. The simulation studies that we have conducted, however, demonstrate the importance of considering multiple parameter initializations in order to fully maximize model likelihoods. These simulations provide a conservative estimate of how frequently our algorithm results in improved likelihoods compared to existing standards.

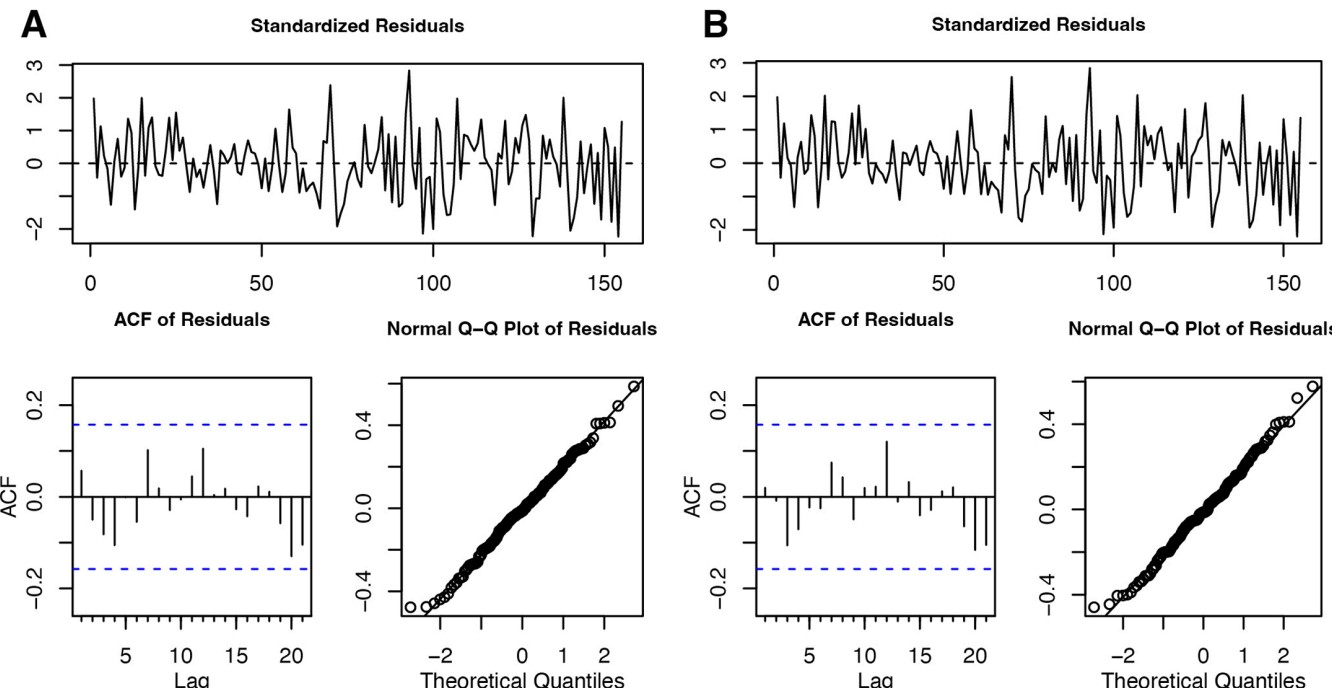

**Fig 8. Residual plots for models fit to the Lake Michigan-Huron data.** (A) Residuals of fitted AR(1) model. (B) Residuals of fitted ARMA(2,1) model.

A common situation where our algorithm is expected to provide even larger improvements than those reported here is in the presence of missing data, a primary motivator of the likelihood maximization procedure of existing software [14]. In this situation, the well-informed CSS initialization is not available, and the default approach is to initialize at the origin, resulting in a greater need to attempt multiple parameter initializations.

Parameter estimates corresponding to higher likelihood values are not necessarily scientifically preferable to alternative regions of parameter space with lower likelihood values [26]. Sometimes, our improved estimates may result in models with nearly canceling roots, parameters near boundary conditions, or otherwise unfavorable statistical properties. On other occasions, our method can rescue a naive optimization attempt from a local maximum having those unfavorable properties. Alternative forms of parameter estimation may also be preferable in some cases. For example, a series of papers by Chen and Deo show that restricted likelihood estimators have favorable properties compared to maximum likelihood estimators when the AR parameters are near the unit root [8–10]. Practitioners should carefully evaluate fitted models to ensure they are appropriate for the data and problem at hand.

The primary limitation of our approach is that it achieves higher likelihoods at the cost of processing speed, which is more pronounced with large datasets. However, our algorithm is most necessary for small datasets ($n \ll 10000$), where default parameter initialization strategies may perform poorly. Therefore, our algorithm is most beneficial for small to moderate sample sizes, where the additional computational cost is generally negligible. The compute time of our algorithm is approximately $K$ times slower than the default approach, where $K$ is the number of unique parameter initializations. This is only an approximation of the actual additional cost as not all initializations require the same amount of processing time in

order to converge. In particular, initializations that are already close to local maximum will generally converge much quicker than those that are further away.

Our proposed algorithm for ARMA parameter estimation significantly advances statistical practice by addressing a frequently occurring optimization deficiency. Because existing software can also be leveraged to mitigate the issue, the largest contribution of this work may be highlighting the prevalence of this optimization problem. Traditional random initialization approaches software fail to uniformly cover the entire range of possible models and often produce many initializations outside the accepted range. Our algorithm offers a computationally efficient and practically convenient solution, providing a robust approach to parameter initialization and estimation that ensures adequate coverage of all possible models. We have shown that it provides a new standard for best practice in the field of time series analysis.

## Supporting information

**S1 Appendix. Uniform sampling.** An appendix demonstrating why uniform sampling of ARMA model parameters is impractical.
(PDF)

**S2 Appendix. Python comparison.** An appendix comparing our algorithm to existing Python software.
(PDF)

**S3 File. Source code.** All code used to run the simulations studies used in this article and supplement, zipped into a single file.
(ZIP)

## Author contributions

**Conceptualization:** Jesse Wheeler, Edward L. Ionides.

**Formal analysis:** Jesse Wheeler.

**Investigation:** Jesse Wheeler.

**Methodology:** Jesse Wheeler.

**Project administration:** Jesse Wheeler.

**Software:** Jesse Wheeler.

**Supervision:** Edward L. Ionides.

**Visualization:** Jesse Wheeler.

**Writing – original draft:** Jesse Wheeler.

**Writing – review & editing:** Jesse Wheeler, Edward L. Ionides.

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
