## [Decision Letter · Decision Letter 0]

23 Jun 2025

PONE-D-25-22432Revisiting Inference for ARMA Models: Improved Fits and Superior Confidence IntervalsPLOS ONE

Dear Dr. Wheeler,

Thank you for submitting your manuscript to PLOS ONE. After careful consideration, we feel that it has merit but does not fully meet PLOS ONE’s publication criteria as it currently stands. Therefore, we invite you to submit a revised version of the manuscript that addresses the points raised during the review process.

We look forward to receiving your revised manuscript.

Kind regards,

Mohamed R. Abonazel, Ph.D.

Academic Editor

PLOS ONE

Journal Requirements:

2. We are unable to open your Supporting Information files [sourceFiles.zip, arima2_3.3.0.tar.gz] Please kindly revise as necessary and re-upload.

Reviewer's Responses to Questions

**Comments to the Author**

1. Is the manuscript technically sound, and do the data support the conclusions?

Reviewer #1: Yes

Reviewer #2: Yes

Reviewer #3: Yes

Reviewer #4: Yes

2. Has the statistical analysis been performed appropriately and rigorously? 

Reviewer #1: Yes

Reviewer #2: Yes

Reviewer #3: Yes

Reviewer #4: Yes

3. Have the authors made all data underlying the findings in their manuscript fully available?

Reviewer #1: Yes

Reviewer #2: Yes

Reviewer #3: Yes

Reviewer #4: No

4. Is the manuscript presented in an intelligible fashion and written in standard English?

Reviewer #1: Yes

Reviewer #2: Yes

Reviewer #3: Yes

Reviewer #4: Yes

5. Review Comments to the Author

Reviewer #1: The manuscript is clear and sound all the graphs and results are properly explained although the format of the paper is unusual you should specicify the literature review section also follow the proper layout of a research paper.Section 1 introduction section 2 literature review 3 methodoogy 4 results and discussions 5 Conclusion then followed by your appendix

Reviewer #2: The paper is very nice and tries to address an important and not adequately recognized problem. However, the solution provided by the authors is ad-hoc. That said, there is an available result in the literature that allows the authors to address this criticism and a revised paper will be suitable for publication.

Reviewer #3: Review for PONE-D-25-22432 “Revisiting Inference for ARMA Models: Improved Fits and Superior Confidence Intervals” by Jesse Wheeler and Edward

L. Ionides

This manuscript makes a significant contribution to time series analysis by addressing a

critical flaw in ARMA model estimation, proposing a novel multi-start algorithm to improve

parameter estimation, and demonstrating the superiority of profile likelihood confidence

intervals over those based on Fisher’s information matrix. The simulation studies and realworld example provide robust evidence of the method’s efficacy, and the work is implemented

in the R package arima2, enhancing its practical utility. Overall, the paper is well-written,

and the topic aligns well with the journal’s scope. However, the manuscript requires several

clarifications. I therefore recommend acceptance once the comments below are appropriately

addressed.

1. In Section 2.2, on Page 6, lines 196–198, the authors mention that 36,000 unique

time series were generated. Does this mean that 1,000 independent time series were

generated for each combination of n, p, and q? Were the model parameters fixed or

randomly generated? A brief clarification in the manuscript would be helpful.

2. In Figure 1, the authors demonstrate optimization issues—i.e., sub-optimal parameter

estimates—in four generated MA(1) models. However, AR(p) models for p = 1, 2, 3

and MA(q) models for q = 1, 2, 3 are not included in the simulation results shown in

Figure 2. Could the authors consider adding these models to Figure 2 to present a

more complete set of ARMA(p, q) models with p, q ≤ 3?

3. In Section 2.2, on Page 7, lines 228–229, the authors mention that the average computation time for obtaining the MLEs was 0.6 seconds using the proposed method.

What is the corresponding average computation time using the standard approach for

ARMA parameter estimation? Including a table or figure comparing the runtimes of

stats::arima versus arima2::arima across different sample sizes would enhance the

practical relevance of the discussion. Additionally, providing the hardware specifications (e.g., processor, RAM) would help clarify the computational context.

4. In Figure 3A, were the confidence intervals based on Fisher’s information matrix

constructed using the standard approach or the proposed method? Additionally, does

the arima2 package provide a function for computing profile likelihood confidence

intervals (PLCIs)?

5. In Table 2, were the ARMA(2,1) parameters estimated using the standard approach

or the proposed method? Does arima2::arima include an invertibility check?

1

6. The caption for Table 1 should be moved above the table to be consistent with Table

2.

7. On Page 3, line 92, “do to” should be corrected to “due to”

8. For the annual depths of Lake Michigan example, could the authors include the ACF

and PACF plots of the time series, as well as the ACF plot of the residuals, to aid in

determining the appropriate ARMA orders p and q?

Reviewer #4: The paper "Revisiting Inference for ARMA Models: Improved Fits and Superior

Confidence Intervals" likely deals with improvements to the estimation and inference

procedures for ARMA (AutoRegressive Moving Average) models, focusing on better

model fitting and more accurate/confident interval estimation. Here are some question

should be answered before publication.

Questions:

1. How do the new inference techniques affect the stationarity or invertibility conditions of

ARMA models?

2. How is the accuracy of confidence intervals evaluated?

3. Can the proposed methods be extended to seasonal or multivariate ARMA models (e.g.,

SARIMA, VARMA)?

4. How does this paper's approach compare to bootstrap or Bayesian interval estimation for

ARMA models?

5. Does the paper explore asymptotic properties of the new estimators or intervals (e.g.,

consistency, normality)?

6. Are there improvements in model diagnostics or residual analysis with the new method?

7. Does the method improve convergence in the likelihood optimization process? If yes,

how?

8. What improvements are observed in the width, coverage, or accuracy of these intervals

compared to traditional asymptotic intervals?

9. How does the paper ensure that improved fits don’t result in overfitting or spurious

models?

10. Are the likelihood surfaces smoother or better-behaved under the new method? Is there

any analysis of local minima or optimization difficulty?

11. What is the purpose of initializing parameters ψ₀ using the CSS (Conditional Sum of

Squares) estimate? Why not start from random values?

12. How does the probability parameter p (used to sample real vs. complex root pairs) affect

the exploration of the parameter space?

13. What role does the bound γ ∈ (0, 0.5) play in root sampling, and how does it ensure

stationarity and invertibility?

14. In case of an odd number of AR or MA roots, why is the non-paired root sampled only at

angle 0 or π?

15. What is the significance of sampling root magnitudes from U(γ, 1−γ)? What would

happen if γ were too close to 0 or 0.5?

16. How sensitive are the results to the choice of parameters α, γ, and p? Should these be

tuned or fixed?

17. Does the improved inference primarily help in estimation, or does it consistently translate

into better predictive performance as well? Were forecasts generated using their algorithm

compared with those from classical ARMA implementations (e.g., Box-Jenkins, Kalman

filter-based methods)?

18. Were the improvements in forecasting found to be statistically significant? Was any test

(e.g., Diebold-Mariano) used to validate forecast accuracy?

19. Were forecasting accuracy metrics — such as RMSE, MAE, MAPE, or out-of-sample

log-likelihood — used in the evaluation? What forecasting horizons (e.g., 1-step, multi-step)

were considered when comparing the improved MLE ARMA with the traditional approach?

6. PLOS authors have the option to publish the peer review history of their article (what does this mean?). If published, this will include your full peer review and any attached files.

Reviewer #1: **Yes: **Muhammad Ahmad

Reviewer #2: No

Reviewer #3: No

Reviewer #4: **Yes: **KAUSTAV ADITYA, ICAR-IASRI, NEW DELHI

---

## [Author Response · Author response to Decision Letter 1]

19 Jul 2025

Thank you for your thoughtful review and consideration of my manuscript. All responses to editor and reviewer comments are provided in an attachment.

---

## [Decision Letter · Decision Letter 1]

15 Aug 2025

PONE-D-25-22432R1Revisiting Inference for ARMA Models: Improved Fits and Superior Confidence IntervalsPLOS ONE

Dear Dr. Wheeler,

Thank you for submitting your manuscript to PLOS ONE. After careful consideration, we feel that it has merit but does not fully meet PLOS ONE’s publication criteria as it currently stands. Therefore, we invite you to submit a revised version of the manuscript that addresses the points raised during the review process.

We look forward to receiving your revised manuscript.

Kind regards,

Mohamed R. Abonazel, Ph.D.

Academic Editor

PLOS ONE

Journal Requirements:

Reviewers' comments:

Reviewer's Responses to Questions

**Comments to the Author**

1. If the authors have adequately addressed your comments raised in a previous round of review and you feel that this manuscript is now acceptable for publication, you may indicate that here to bypass the “Comments to the Author” section, enter your conflict of interest statement in the “Confidential to Editor” section, and submit your "Accept" recommendation.

Reviewer #1: All comments have been addressed

Reviewer #2: All comments have been addressed

Reviewer #3: All comments have been addressed

Reviewer #4: All comments have been addressed

2. Is the manuscript technically sound, and do the data support the conclusions?

Reviewer #1: No

Reviewer #2: Yes

Reviewer #3: (No Response)

Reviewer #4: Yes

3. Has the statistical analysis been performed appropriately and rigorously? 

Reviewer #1: Yes

Reviewer #2: Yes

Reviewer #3: (No Response)

Reviewer #4: Yes

4. Have the authors made all data underlying the findings in their manuscript fully available?

Reviewer #1: Yes

Reviewer #2: Yes

Reviewer #3: (No Response)

Reviewer #4: Yes

5. Is the manuscript presented in an intelligible fashion and written in standard English?

Reviewer #1: Yes

Reviewer #2: Yes

Reviewer #3: (No Response)

Reviewer #4: Yes

6. Review Comments to the Author

Reviewer #1: This research is written in standard english,Also this research is technically sound and all the statistical analysis is done properly.

Reviewer #2: The paper has been adequately revised, in my opinion and is suitable for publication. The authors should consider a deeper investigation of REML in their future work.

Reviewer #3: (No Response)

Reviewer #4: 1. While processing time is discussed, more formal benchmarking across software environments (e.g., R vs Python) would help practitioners decide when to use these methods.

2.The focus is primarily on Gaussian noise. The authors mention non-Gaussian models in passing—some brief results or commentary on how the proposed methods extend to robust or non-Gaussian ARMA models would enhance generalizability.

3. The choice of stopping criterion (no new maximum in M steps) is somewhat heuristic. A more adaptive or data-driven rule, or guidance on setting M based on sample size or model order, would be helpful.

4.Figures mentioned (e.g., Figs 1–8) are informative, but it would be useful to annotate likelihood surfaces more explicitly in plots to show how initializations evolve toward the global maximum.

5. Though the method improves likelihoods, a brief reflection on whether overfitting could result (especially in small samples with complex models) would strengthen the discussion.

6.The manuscript is long but well-organized. Still, trimming some of the implementation details from the main text to Supplementary Material could help readability.

7.Cite and contrast with more recent Bayesian optimization or machine learning-based approaches to global optimization for time series models.

This paper addresses a subtle but widespread flaw in statistical practice. The proposed solution is innovative, reproducible, and impactful. With small refinements, it can become a go-to reference for robust ARMA inference. the paper can be accepted with minor revision.

7. PLOS authors have the option to publish the peer review history of their article (what does this mean?). If published, this will include your full peer review and any attached files.

Reviewer #1: No

Reviewer #2: No

Reviewer #3: No

Reviewer #4: **Yes: **Dr. Kaustav Aditya, ICAR-IASRI, New Delhi

---

## [Author Response · Author response to Decision Letter 2]

8 Sep 2025

We have included our response to the editor and reviewer comments as an attachment in our submission.

---

## [Editor Report · Decision Letter 2]

22 Sep 2025

Revisiting Inference for ARMA Models: Improved Fits and Superior Confidence Intervals

PONE-D-25-22432R2

Dear Dr. Wheeler,

We’re pleased to inform you that your manuscript has been judged scientifically suitable for publication and will be formally accepted for publication once it meets all outstanding technical requirements.

Kind regards,

Mohamed R. Abonazel, Ph.D.

Academic Editor

PLOS ONE

---

## [Editor Report · Acceptance letter]

PONE-D-25-22432R2

PLOS ONE

Dear Dr. Wheeler,

I'm pleased to inform you that your manuscript has been deemed suitable for publication in PLOS ONE. Congratulations! Your manuscript is now being handed over to our production team.

Kind regards,

on behalf of

Dr Mohamed R. Abonazel

Academic Editor

PLOS ONE